# Mechanistic Origin of Different Binding Affinities of SARS-CoV and SARS-CoV-2 Spike RBDs to Human ACE2

**DOI:** 10.3390/cells11081274

**Published:** 2022-04-09

**Authors:** Zhi-Bi Zhang, Yuan-Ling Xia, Jian-Xin Shen, Wen-Wen Du, Yun-Xin Fu, Shu-Qun Liu

**Affiliations:** 1State Key Laboratory for Conservation and Utilization of Bio-Resources in Yunnan, School of Life Sciences, Yunnan University, Kunming 650091, China; zhangzhibi@kmmu.edu.cn (Z.-B.Z.); xiayl@ynu.edu.cn (Y.-L.X.); Jianxins0806@163.com (J.-X.S.); wenwendu2014@hotmail.com (W.-W.D.); 2Yunnan Key Laboratory of Stem Cell and Regenerative Medicine and Biomedical Engineering Research Center, Kunming Medical University, Kunming 650500, China; 3Human Genetics Center and Department of Biostatistics and Data Science, School of Public Health, The University of Texas Health Science Center, Houston, TX 77030, USA

**Keywords:** molecular dynamics, binding free energy calculations, electrostatic interactions, amino acid residue changes, protein-protein binding affinity, binding interfaces

## Abstract

The receptor-binding domain (RBD) of the SARS-CoV-2 spike protein (RBD_CoV2_) has a higher binding affinity to the human receptor angiotensin-converting enzyme 2 (ACE2) than the SARS-CoV RBD (RBD_CoV_). Here, we performed molecular dynamics (MD) simulations, binding free energy (BFE) calculations, and interface residue contact network (IRCN) analysis to explore the mechanistic origin of different ACE2-binding affinities of the two RBDs. The results demonstrate that, when compared to the RBD_Co__V2_-ACE2 complex, RBD_Co__V_-ACE2 features enhanced dynamicsand inter-protein positional movements and increased conformational entropy and conformational diversity. Although the inter-protein electrostatic attractive interactions are the primary determinant for the high ACE2-binding affinities of both RBDs, the significantly enhanced electrostatic attractive interactions between ACE2 and RBD_Co__V2_ determine the higher ACE2-binding affinity of RBD_CoV2_ than of RBD_CoV_. Comprehensive comparative analyses of the residue BFE components and IRCNs between the two complexes reveal that it is the residue changes at the RBD interface that lead to the overall stronger inter-protein electrostatic attractive force in RBD_CoV2_-ACE2, which not only tightens the interface packing and suppresses the dynamics of RBD_Co__V2_-ACE2, but also enhances the ACE2-binding affinity of RBD_Co__V2_. Since the RBD residue changes involving gain/loss of the positively/negatively charged residues can greatly enhance the binding affinity, special attention should be paid to the SARS-CoV-2 variants carrying such mutations, particularly those near or at the binding interfaces with the potential to form hydrogen bonds and/or salt bridges with ACE2.

## 1. Introduction

The severe acute respiratory syndrome coronavirus 2 (SARS-CoV-2) [1,2], which shares a high homology with SARS-CoV (about 80% identical at the genome level) in the 2002–2003 outbreak [3], causes the ongoing coronavirus disease 2019 (COVID-19). As of January 2022, SARS-CoV-2 has caused over 480 million cases of COVID-19 and more than 6.1 million deaths worldwide (https://www.who.int/emergencies/diseases/novel-coronavirus-2019 (accessed on 29 March 2022)).

Both SARS-CoV and SARS-CoV-2 belong to the *Sarbecovirus* subgenus of betacoronaviruses [4] and utilize the same receptor, angiotensin-converting enzyme 2 (ACE2) for cell entry. The infection process is triggered by the attachment of the CoV spike protein to the cell-surface receptor [5,6]. The CoV spike protein is a class I membrane fusion glycoprotein [7] composed of three identical protomers protruding from the surface of lipid-enveloped CoV particles [8,9]. Each protomer consists of two subunits, S1 and S2, which are post-translationally cleaved products from the single-chain polypeptide spike precursor. Furthermore, S1 and S2 are responsible for virus attachment to cells and for the fusion of the viral and cellular membranes, respectively [7,10,11,12]. In the spike trimer, the N- and C-terminal portions of a S1 subunit fold independently as two large domains, known as the N-terminal domain (NTD) and C-terminal domain (CTD). The latter serves as the receptor-binding domain (RBD) in SARS-CoV and SARS-CoV-2 [11,13,14,15,16,17]. The S1 subunit, especially the RBD, is also the immunodominant target of the humoral response, and hence, is the uppermost component of both mRNA and adenovirus-based vaccines [18,19].

The spike trimer does not present as a single rigid conformation but rather as an ensemble of different conformations/states (i.e., the closed state with all RBDs in the down orientation and the open states with one, two, or three erect RBD(s)) that coexist in equilibrium with different population distributions [16,17,20]. In the closed conformation, the ACE2-binding surface (i.e., partial surface of the receptor binding motif (RBM)) on RBDs is buried inside the trimer, inaccessible to the ACE2 because of the down orientations of the three RBDs and their tight packing against one another [16]. The closed state can spontaneously convert to the open states through a hinge-like motion that progressively lifts RBDs up, thus allowing binding to the ACE2 due to the full exposure of RBM [21]. Upon binding, the first ACE2-bound RBD is stabilized in the up orientation, and this promotes the other two RBDs to consecutively lift up and bind to the ACE2 until reaching the fully open, three-ACE2-bound conformation, which is responsible for priming the spike trimer for S2 unsheathing and the following membrane fusion [17].

Although the erection of the RBD is a prerequisite for ACE2 binding, the RBD is an independently folded domain and its erection has only a marginal impact on its overall conformation [17]. As a result, the binding affinity of the spike protein to the ACE2 was usually evaluated using the RBD rather than the spike trimer, although previous computational studies showed that certain mutations outside the SARS-CoV-2 spike RBD were capable of influencing an affinity to the ACE2 through altering the spike conformational dynamics [22,23]. Interestingly, multiple earlier experimental and simulation studies have collectively demonstrated that the SARS-CoV-2 RBD (RBD_CoV2_) has a higher ACE2-binding affinity than the SARS-CoV RBD (RBD_CoV_) [16,24,25,26,27,28,29]. To a large extent this explains why SARS-CoV-2 manifests an increased infectivity and transmissibility in comparison to SARS-CoV.

The crystal structures of RBD_CoV_ and RBD_CoV2_ in complex with human ACE2 have been determined at high resolution [13,14], revealing that the two RBDs share not only overall similar conformations (with C_α_ root mean square deviation (RMSD) of 0.47 Å) but also nearly identical modes of binding to ACE2 (Figure 1A–C). Both RBDs have two subdomains: a core and a RBM. The former is a twisted five-stranded antiparallel β-sheet connected by short helices and loops, while the latter is composed of a short two-stranded antiparallel β-sheet, two short helices, and several long loops. The core contains few residues capable of making contacts with ACE2, and most of the ACE2-contacting residues are from RBM. The sequence identity shared by the RBD_CoV_ and RBD_CoV2_ is 73.2%, which explains why they have highly similar overall structures. However, the sequence identity of the cores increases to 88.0%, and that of the RBMs falls to 47.8%. This drastic contrast may partially explain the difference between the ACE2-binding affinities of the two RBDs, as being due to more residue changes and much more ACE2-contacting residues in the RBM than in the core (Figure 1D).

Although multiple previous studies have provided insight into the structural and molecular basis responsible for the different binding affinities of RBD_CoV_ and RBD_CoV2_ to human ACE2 [24,25,30,31,32,33], the underlying mechanisms modulating the mechanics and energetics of RBD-ACE2 interactions still remain to be elucidated. In order to explore the mechanistic reasons for the experimentally observed difference in the ACE2-binding affinities of the two RBDs, we performed multiple-replica MD simulations on the structures of the RBD_Co__V_-ACE2 and RBD_Co__V2_-ACE2 complexes. The study included comparative analyses in terms of dynamics and thermodynamics, calculations of the protein-protein and per-residue binding free energies (BFEs), constructions of the interface residue contact networks (IRCNs), and comprehensive comparative analyses of IRCNs, interface interactions, and BFE components of individual residues. The new results shed light on the dynamics and energetic aspects of the modulation mechanisms of RBD-ACE2 interactions, and they further explain why RBD_C__o__V__2_ enhances its ACE2-binding affinity compared with RBD_C__o__V_, which may help the surveillance of novel SARS-CoV-2 variants.

## 2. Materials and Methods

### 2.1. Structural Preparation

The X-ray crystallographic structures of the RBD_CoV_-ACE2 and RBD_CoV2_-ACE2 complexes were obtained from the Protein Data Bank (PDB; http://www.rcsb.org (accessed on 8 August 2021)) with PDB IDs 2AJF [13] and 6M0J [14], respectively, which were chosen because of their high resolutions (2.90 and 2.45 Å, respectively) and the wild-type (WT) binding partners in both complex structures. The hetero atoms and crystallographic water molecules were removed, whereas the atomic coordinates for ACE2, RBD, and ACE2-bound Zn^2+^ and Cl^−^ ions were retained. For the RBD_CoV_-ACE2 complex, the missing atomic coordinates of the RBD_CoV_ residues 320-322, 376-381, and 503-512 were modeled using the SWISS-MODEL server [34] with the RBD_CoV2_ structure as the template. For the two RBDs, the structure-based sequence alignment was performed using the Dali server (http://ekhidna.biocenter.helsinki.fi/dali/ (accessed on 12 August 2021)) [35] and visualized by ESPript 3.0 (https://espript.ibcp.fr/ESPript/cgi-bin/ESPript.cgi (accessed on 12 August 2021)) [36]. RBD_CoV2_ numbering was used throughout for simplicity.

### 2.2. MD Simulations

All MD simulations were performed using the GROMACS 5.1.4 software package [37]. The AMBER99SB-ILDN force field [38] was employed because its performance is usually considered among the best of currently available all-atom force fields for protein MD simulations due to the improvement in side-chain torsion potentials of amino acids [38,39,40,41]. The p*K*_a_ values of all titratable residues were calculated using the PropKa server (https://www.ddl.unimi.it/vegaol/propka.htm (accessed on 15 August 2021)) [42] to assign their protonation states at pH 7.4. The predicted p*K*_a_ values of all lysines and arginines and all glutamate and aspartate residues are greater and less than 7.4, respectively; therefore, they were protonated (positively charged) and deprotonated (negatively charged), respectively. Since His374 of ACE2 (hereafter referred to as ACE2:His374) in both complexes and ACE2:His493 in RBD_CoV2_-ACE2 have predicted p*K*_a_ values greater than 7.4, their imidazole N_δ__1_ and N_ε__2_ atoms were both protonated (positively charged). All the other histidines have p*K*_a_ values less than 7.4, and hence, were treated as the uncharged neutral ones, with their imidazole rings being protonated automatically on either N_δ1_ or N_ε2_ based on the optimal hydrogen-bonding conformation using the GROMACS tool ‘gmx pdb2gmx’.

After protonation, each complex structure was solvated with the TIP3P water model [43] in a periodic dodecahedron box with a minimum solute-box wall distance of 1.2 nm. The net charges of both systems were neutralized with a 0.15 M concentration of NaCl to mimic the physiological conditions. Each system was subjected to the steepest descent of energy minimization until no significant energy change could be detected. To effectively ‘soak’ the solute into the solvent, four consecutive 100-ps NVT MD simulations were conducted at 310 K, with the protein heavy atoms restrained by decreasing harmonic potential force constants of 1000, 100, 10, and 0 kJ/mol/nm^2^. To improve the sampling of the conformational space, each system was subjected to 10 independent 15-ns production MD simulations, with each replica initialized with different initial atomic velocities assigned from a Maxwell distribution at 310 K. In the production MD runs, the LINear Constraint Solver (LINCS) algorithm [44] was used to constrain all bonds to equilibrium lengths, thus allowing a time step of 2 fs. The particle-mesh Ewald (PME) method [45] was used to treat the long-range electrostatic interactions, with a real-space cut-off of 1.0 nm, grid spacing of 0.12 nm, and interpolation order of 4. The van der Waals (vdW) interactions were treated using the Verlet scheme with a cut-off distance of 1.0 nm. The system temperature was controlled at 310 K using the velocity-rescaling thermostat [46] with a time constant of 0.1 ps. The system pressure was maintained at 1 atm using the Parrinello–Rahman barostat [47], with a time constant of 2.0 ps. System coordinates were saved every 2 ps.

### 2.3. MD Trajectory Analysis

For the 10 replica trajectories of each complex, the time dependent C_α_ RMSD values, relative to the starting structure, were calculated using the GROMACS tool ‘gmx rms’. For each complex, the equilibrated trajectory portion for each of the 10 replicas was concatenated into a single joined equilibrium trajectory using the GROMACS tool ‘gmx trjcat’. The principal component analysis (PCA) was performed on the C_α_ atoms of the single joined equilibrium trajectory using the GROMACS tools ‘gmx covar’ and ‘gmx anaeig’. The selection of the C_α_, rather than the backbone atoms, is sufficient to provide reliable information about the large concerted protein motions during simulations while reducing the computation cost. Then, the first two eigenvector projections were used as the reaction coordinates to reconstruct the two-dimensional free energy landscape (FEL) [48,49,50] with the following equation:*F*(*s*) = −*k*_B_*T*ln(*P*_i_/*P*_max_)(1)
where *k*_B_ is the Boltzmann’s constant, *T* is the simulation temperature, *P*_i_ is the probability of finding the system in state i that is characterized by the two reaction coordinates (i.e., the first two eigenvectors), and *P*_max_ is the probability of the most probable state. An in-house Python script (Appendix A) was used to implement the above model for generating FELs of the two complexes.

For each complex, 100 conformations were randomly extracted from the global free energy minimum of FEL and were treated as the representative structures for the subsequent BFE calculation and interaction/contact network analysis.

### 2.4. Binding Free Energy Calculation

The BFEs of RBD_CoV_-ACE2 and RBD_CoV2_-ACE2 were evaluated using the molecular mechanics Possion–Boltzmann surface area (MM-PBSA) method, as implemented in the GROMACS tool g_mmpbsa [51]. MM-PBSA is an endpoint approach capable of estimating the protein-protein/ligand BFE based merely on the structure or structural ensemble of the bound complex without considering either the physical or the non-physical intermediates [52]. Here, the BFE was estimated with the single trajectory approach, which assumes that the conformations of the free RBD and free ACE2 are identical to those in the protein–protein complex.

In MM-PBSA, the BFE (Δ*G*_binding_) of a single protein–protein complex was calculated as follows:Δ*G*_binding_ = Δ*E*_MM_ + Δ*G*_solvation_ − *T*Δ*S =* (Δ*E*_bonded_
*+* Δ*E*_vdW_ + Δ*E*_elec_) + (Δ*G*_polar_ + Δ*G*_non-polar_) − *T*Δ*S*(2)
where Δ*E*_MM_, Δ*G*_solvation_, and *T*Δ*S* are the changes in the vacuum molecular mechanics potential energy, solvation free energy, and solute entropy, respectively, upon the complex formation of two proteins. Δ*E*_MM_ is further decomposed into the bonded energy change (Δ*E*_bonded_) and changes in the vdW (Δ*E*_vdW_) and electrostatic (Δ*E*_elec_) potential energies. The Δ*E*_bonded_ value is zero due to the single trajectory approach, and △*E*_vdW_ and Δ*E*_elec_ are the vdW and electrostatic interaction energies between the two binding partners, respectively. Δ*G*_solvation_ is decomposed into the polar (Δ*G*_polar_) and non-polar (Δ*G*_non-polar_) solvation free energy contributions, with Δ*G*_polar_ actually representing the electrostatic desolvation of free energy, and Δ*G*_non-polar_ representing the hydrophobic effect arising from the solvent entropy gain upon binding [52,53,54]. The term *T*Δ*S*, which is often approximated using the normal mode method [55], was not included in our calculations for two reasons: the first is that the estimation of the solute entropy change is a time-consuming task, often resulting in highly uncertain results with a larger standard error than those of the other energy terms [56,57,58], and the second reason is that omitting the solute entropy term likely has only a minor impact on the comparison between the relative BFEs of different ligands/proteins to the same receptor protein [51,56]. The calculated BFE values in our work are the ones of the relative binding energies, which, although unphysical due to the neglect of the solute entropy change, can be used to compare binding affinities of different ligands to a common receptor.

For each complex, MM-PBSA calculations were performed on a structural ensemble of 100 representative conformations. The g_mmpbsa default parameters were used with the exceptions for calculating the polar solvation free energy (*G*_polar_). *G*_polar_ was calculated by solving the nonlinear PB equation (PBsolver = npbe) with the following settings: (i) grid resolution (gridspace) of 0.4 Å, (ii) ionic strength (NaCl concentration) of 0.15 M with radii for Na^+^ and Cl^−^ of 0.95 and 1.81 Å, respectively, (iii) dielectric constants of the solute (pdie), solvent (sdie), and vacuum (vdie) of 4, 80, and 1, respectively, and (iv) temperature (temp) of 310 K.

Per-residue contribution to the total BFE (hereafter referred to as the residue BFE) was obtained by implementing the ‘binding energy decomposition’ module of g_mmpbsa. This module decomposes the total BFE into contributions from individual residues by calculating each atom’s energy components (i.e., *E*_MM_*, G*_polar_*,* and *G*_nonpolar_) of a residue in both the free and bound forms.

### 2.5. Interface Interaction/Contact Analyses

The IRCN was constructed based on the numbers of close inter-atomic contacts of involved residues from RBD and ACE2. A close inter-atomic contact is considered to exist if the “overlap” value between any two atoms is greater than −4.0 Å. The overlap between atoms i and j (*overlap*_ij_) is defined as:*overlap*_ij_ = *r*_vdWi_ + *r*_vdWj_ − *d*_ij_(3)
where *r*_vdWi_ and *r*_vdWj_ represent the vdW radii of atoms i and j, respectively, and *d*_ij_ denotes the distance between their nuclei. A hydrogen bond (HB) is considered to exist when the distance between the donor and acceptor atoms is less than 3.5 Å and the angle from the donor atom over the hydrogen to the acceptor atom is greater than 120°. A salt bridge (SB) is considered to be formed if the distance between any side-chain oxygen atom from a negatively charged residue (Asp or Glu) and any side-chain nitrogen atom from a positively charged residue (Arg or Lys) is less than 4.0 Å.

For each representative structure of the RBD_Co__V_-ACE2 and RBD_Co__V2_-ACE2 complexes, close inter-atomic contacts, HBs, and SBs were identified using the Chimera ‘Find Contacts’ tool [59] and VMD plugins ‘Hydrogen Bonds’ and ‘Salt Bridge’ [60], respectively. The average number of close inter-atomic contacts in a given contacting residue pair was calculated over the 100 representative structures; only the residues with an average contact number greater than 1.0 were considered as the binding interface residues. The occupancy of a HB or SB was calculated as the fraction of the structures, within which a specified HB/SB exists out of the 100 representative structures; only those with occupancy greater than 20% were considered as the stable HBs/SBs. Finally, the IRCN and interface HB were generated from the 100 representative structures of each complex by Chimera 1.14 [59] and visualized by Cytoscape 3.8.1 [61]. The distance of a given residue to the binding interfaces, which is defined as the minimum distance between any pair of atoms from the given residue and from the interface residues, was calculated over the 100 representative structures using the GROMACS tools ‘gmx pairdist’.

## 3. Results

### 3.1. Structural Fluctuations during Simulations

Figure 2 shows the time dependent C_α_ RMSD values of the two complexes, relative to their respective starting structures during the multiple-replica simulations. All 10 replicas of RBD_CoV2_-ACE2 require only a few ps to reach relatively stable RMSD values (Figure 2B). Whereas some replicas of RBD_CoV_-ACE2 require more than 1.3 ns to reach a relative plateau of RMSD values (Figure 2A). In order to ensure the temporal consistency of the two simulation systems, we arbitrarily treated the 2–15-ns trajectory of each replica as the equilibrated portion. It is clear that (i) the equilibrated portions of the 10 RBD_CoV_-ACE2 replicas span a wider RMSD range (0.14–0.49 nm) than those of the 10 RBD_CoV2_-ACE2 replicas (0.11–0.31 nm) and, (ii) RBD_CoV_-ACE2 has more replicas with a fluctuation amplitude greater than 0.15 nm than RBD_CoV2_-ACE2. These observations indicate that during equilibrium, RBD_CoV_-ACE2 deviated more from its starting conformation and experienced larger global structural fluctuations than RBD_CoV2_-ACE2. Nevertheless, visual inspection of all replica trajectories revealed that both RBD_CoV_ and RBD_CoV2_ remained stably associated with ACE2 throughout the 15-ns simulations. 

In order to further evaluate the structural fluctuations of the two binding partners and their relative mobility with respect to each other, we calculated the time dependent C_α_ RMSD values of the RBD and ACE2 using two ways of the least-squares fitting (i.e., fitting to their respective structures (self-fitting) and to the structure of the other partner (non-self-fitting) in the starting complex). For both complexes, the self-fitting RMSD values of RBDs (Appendix A) and ACE2s (Appendix A) are clearly lower than their respective non-self-fitting values (Appendix A). However, the self-fitting RMSD curves for RBD and ACE2 span a wider width and have a larger fluctuation amplitude in RBD_Co__V_-ACE2 (Appendix A) than in RBD_Co__V2_-ACE2 (Appendix A), respectively. This implies that the two binding partners have a larger structural variability in the former complex. Interestingly, for both complexes, although their non-self-fitting RMSD curves of RBD and ACE2 collectively show drastic fluctuations, the obviously wider curve widths observed for RBD_CoV_-ACE2 indicate that its two binding partners experienced larger relative positional movements than the two partners in RBD_CoV2_-ACE2.

Taken together, when compared to the RBD_CoV2_-ACE2 complex, RBD_CoV_-ACE2 experienced not only more drastic structural fluctuations at both the levels of the entire complex and individual binding partners, but also larger relative positional movements between the two partners. Thus, RBD_CoV_-ACE2 has a lower structural stability and a less stable binding orientation between the two partners than RBD_CoV2_-ACE2. 

### 3.2. Principal Components and Free Energy Landscapes

For each complex, PCA analysis was performed on its single joined equilibrium trajectory to extract the most important PCs or eigenvectors. For both complexes, the first two eigenvectors possess the largest eigenvalues (Appendix A), and their cumulative eigenvalues contribute 62.0% and 48.5% to the total mean square fluctuation values of RBD_CoV_-ACE2 and RBD_CoV__2_-ACE2, respectively (Appendix A, inset). Since the conformational space is spanned by 3N (N is the number of C_α_ atoms; 790 and 791 in RBD_CoV_-ACE2 and RBD_CoV2_-ACE2, respectively) eigenvectors, it is reasonable to believe that the first two eigenvectors contribute substantially to the overall conformational freedom in the space; therefore, the essential subspace formed by the first two eigenvectors contains the main conformational states/substates sampled by the MD simulations, from which the most representative conformations can be identified through reconstructing the FEL.

Figure 3 shows the two-dimensional FELs of the two complexes. It is clear that the FEL of RBD_Co__V_-ACE2 (Figure 3A) covers a larger region in the essential subspace than RBD_Co__V2_-ACE2’s FEL (Figure 3B), indicating a larger conformational entropy of the former complex. Furthermore, the FEL of RBD_Co__V_-ACE2 features a rough/rugged surface because it contains two large basins (with a free energy level lower than −10 kJ/mol), within which multiple local minima (with a free energy level lower than −11 or −12 kJ/mol) are presented. The FEL of RBD_Co__V2_-ACE2 shows a typical funnel-like shape characterized by a continuous falling of free energy until reaching the global free energy minimum (−14 kJ/mol), but without local minima observed on the funnel wall; therefore, it may be considered that during simulations, RBD_Co__V_-ACE2 and RBD_Co__V2_-ACE2 sampled two and one conformational states, respectively, with the former’s two states containing multiple metastable substates and the latter’s single state (in the global minimum) being its most stable state. Nevertheless, the global free energy minima in both FELs have the same free energy level (−14 kJ/mol), which is indicative of the equivalent thermostability of the two complexes. Interestingly, the global minimum has a larger size in RBD_Co__V2_-ACE2’s FEL than in RBD_CoV_-ACE2’s FEL, which is indicative of a larger population of the most thermostable conformations sampled by RBD_CoV2_-ACE2.

In summary, although RBD_Co__V_-ACE2 has a larger conformational entropy and richer conformational diversity than RBD_Co__V2_-ACE2, these two complexes present the equivalent thermostability; therefore, it is reasonable to take the most thermostable conformations as the representative structures of the two complexes. As a result, for each complex, 100 conformations/structures were randomly extracted from the global free energy minimum for the subsequent BFE calculation and IRCN construction.

### 3.3. Binding Free Energy Calculation

Table 1 shows the calculated average values and corresponding standard deviations (SDs) of the BFE components for the two complexes. The average values of the total BFE (ΔG_binding_) for RBD_Co__V_-ACE2 and RBD_Co__V2_-ACE2 are −2289.2 and −2455.0 kJ/mol, respectively. Although the ranges of the Δ*G*_binding_ values of the two complexes are overlapping when taking the SDs into account, the *p*-value of 2.1 × 10^−19^ by the one-sided *t*-test indicates that the BFE of RBD_Co__V2_-ACE2 is statistically significantly lower than that of RBD_Co__V_-ACE2, which is consistent with previous experimental observations demonstrating that the RBD_CoV2_ has a higher ACE2-binding affinity than RBD_CoV_ [16,24,28,29].

For both complexes, the electrostatic interaction potential energy (Δ*E*_elec_) contributes most significantly to lowering BFE, so that this term alone can overcompensate for the large negative contribution from the electrostatic desolvation energy term (Δ*G*_polar_). The difference in the average values of Δ*G*_binding_ from RBD_Co__V_-ACE2 to RBD_Co__V2_-ACE2 is −165.8 kJ/mol. The differences in the average values of Δ*E*_vdW_, Δ*E*_elec_, Δ*G*_polar_, and Δ*G*_non-polar_ are −15.9, −258.6, 109.6, and −1.0 kJ/mol, respectively. Therefore, the higher ACE2-binding affinity of RBD_Co__V2_ primarily originates from the considerably stronger inter-protein electrostatic attractive interactions in RBD_Co__V2_-ACE2 than in RBD_Co__V_-ACE2. Interestingly, RBD_C__o__V_-ACE2 has larger SDs for all the energy terms than RBD_C__o__V__2_-ACE2, indicating a larger dispersion around the respective energy average values in the former complex. This is in agreement with the comparative analysis of the non-self-fitting RMSD values, which reveals a less tight inter-protein association in RBD_CoV_-ACE2 than in RBD_CoV2_-ACE2.

Figure 4A shows the ACE2 residues with the average BFE values greater than 20.0 or lower than −20.0 kJ/mol in both complexes. All these residues are charged ones, with the positively charged and negatively charged residues making the negative and positive contributions, respectively, to the binding affinity of ACE2 to both RBDs. The magnitudes of the BFE values depend on the residue distances to the binding interfaces. The residues located far from the binding interfaces (marked by light or lighter gray rectangles) generally have a smaller magnitude of the absolute values (lower than 40 kJ/mol) than those located near/at the interfaces (greater than 40 kJ/mol). Figure 4C shows the per-residue BFE difference from the RBD_CoV_-bound ACE2 to RBD_CoV2_-bound ACE2. All the residues with significant energy differences (greater than 20 or lower than −20 kJ/mol) are charged ones and reside near/at the binding interfaces with the exception of H493. Notably, His493 has a net charge of 0 and +1 in the RBD_CoV_-bound and RBD_CoV2_-bound ACE2s (for details; see Section 2.2), respectively. Despite its remote distance to interfaces (greater than 3.0 nm; see Figure 4A), the change in the charge property of His493 leads to a considerable difference in its BFE. This indicates the importance of the long-range electrostatic interactions in affecting a residue’s contribution to the binding affinity. For the charged residues located near/at the interfaces, the large changes in the BFE also arise from the differences in the electrostatic interactions. For example, D30 contributes to enhancing ACE2’s affinity to RBD_CoV2_ due to its stronger electrostatic attractive interactions with RBD_CoV2_ (−226.5 kJ/mol; see Appendix A) than with RBD_CoV_ (−66.5 kJ/mol). In contrast, K353 contributes to reducing ACE2’s affinity to RBD_CoV2_, due to its stronger electrostatic repulsive interactions with RBD_CoV2_ (46.6 kJ/mol) than with RBD_CoV_ (9.4 kJ/mol).

On the side of RBDs, the residues with the BFE average values greater than 20 or lower than −20.0 kJ/mol can be either the charged or the uncharged ones (Figure 4B). However, the BFE absolute values of the charged residues (in a range of about 240–550 kJ/mol) are one order of magnitude greater than those of the uncharged residues (in a range of about 24–47 kJ/mol). The positively and negatively charged residues make the positive and negative contributions to the ACE2-binding affinities of both RBDs, respectively, whereas the uncharged residues contribute positively to the ACE2-binding affinities of both RBDs. Interestingly, for the charged residues, those with absolute values greater than 400 kJ/mol are all located at/near the binding interfaces (marked by black/dark gray rectangles), and those with absolute values lower than 300 kJ/mol are located distally from the binding interfaces (gray to lighter gray rectangles). For the uncharged residues, they are all located at the RBD interface (marked by black rectangles).

Figure 4D shows the per-residue BFE difference from RBD_CoV_ to RBD_CoV2_. Clearly, all the significant differences occur during the residue changes involved in the charge changes. The loss of the negatively charged residues (i.e., E354N, D476G, and D494S) and the gain of the positively charged residues (i.e., V417K, T444K, and H458K) result in the negative difference values and contribute to enhancing the ACE2-binding affinity of RBD_CoV2_. The loss of the positively charged residues (i.e., R439N, K452L, K460N, and K478T) and the gain of the negatively charged residues (i.e., V471E and P484E) result in the positive difference values and contribute to reducing the ACE2-binding affinity of RBD_CoV2_. For all the above residue changes, their significant differences in BFE arise from the changes in the electrostatic interaction strength with ACE2 (Appendix A), irrespective of their structural locations. For example, E354N, despite being distal from the binding interfaces, leads to a significant loss in the electrostatic repulsive interactions with ACE2 (305.7 vs. 0.8 kJ/mol), and hence, contributes to enhancing the ACE2-binding affinity of RBD_CoV2_.

In summary, our BFE calculations reveal that (i) the enhanced binding affinity of RBD_CoV2_-ACE2 compared to RBD_CoV_-ACE2 primarily originates from the significantly strengthened inter-protein electrostatic attractive forces in the former complex, (ii) the negatively charged residues in ACE2 and positively charged residues in RBDs make considerable positive contributions to the binding affinity due to their strong electrostatic attractive interactions with the other binding partners, (iii) the positively charged residues in ACE2 and negatively charged residues in RBDs make considerable negative contributions due to their strong electrostatic repulsive interactions, (iv) for the charged residues, their magnitudes of contributions to the binding affinity exhibit a trend of dependence on residue’s distance to the binding interfaces, and (v) the RBD residue changes involved in the charge property changes can greatly impact the ACE2-binding affinities of RBDs through altering the strength of electrostatic interactions with ACE2.

### 3.4. Interface Residue Contact Networks and the Related Interactions

In order to further investigate how the RBD residue changes affect the interface interactions and protein–protein binding affinity, IRCNs were constructed based on the representative structures of the two complexes (Figure 5). The IRCN of RBD_Co__V2_-ACE2 (Figure 5B) contains more nodes and edges than that of RBD_Co__V_-ACE2 (Figure 5A). In addition, there is a higher average number of interface close contacts in RBD_Co__V2_-ACE2 (142.3 ± 20.5) than in RBD_Co__V_-ACE2 (127.0 ± 19.0). These results reveal a tighter interface packing and more intensive interface interactions in RBD_Co__V2_-ACE2 than in RBD_Co__V_-ACE2.

There are eight conserved RBD nodes/residues (dark red) in the two IRCNs (Figure 5A,B). Despite their varying sizes in the two networks, the cumulative number of close contacts made by the conserved nodes is slightly higher in the IRCN of RBD_Co__V_-ACE2 (73.5) than in that of RBD_Co__V2_-ACE2 (68.1). For the conserved RBD residues in the IRCNs of RBD_Co__V_-ACE2 and RBD_Co__V2_-ACE2, the cumulative values of the vdW interaction energy, electrostatic interaction energy, and BFE are −72.8 and −68.9, −102.7 and −94.6, and −119.6 and −109.2 kJ/mol (Appendix A), respectively. These results reveal a trend, that the overall increased number of close contacts on the conserved RBD_Co__V_ residues strengthens their vdW and electrostatic interactions with ACE2; this, in turn, enhances their contribution to the ACE2-binding affinity when compared to the conserved RBD_Co__V2_ residues.

With the exception of RBD:Y/L455, all the non-conserved/changed RBD residues (red nodes) have larger node sizes in RBD_Co__V2_-ACE2’s IRCN than in RBD_CoV_-ACE2’s IRCN. The largest increase in the node size was observed for L486F (residue change from RBD_Co__V_:L486 to RBD_Co__V2_:F486), which enhances the ACE2-binding affinity by −10.2 kJ/mol (Appendix A). Since both RBD_Co__V_:L486 and RBD_Co__V2_:F486 are non-polar amino acids, and they form no HB with ACE2, the enhanced affinity of L486F mainly arises from the increased or additional vdW contacts/interactions with ACE2:M82, Y83, and L79 (Figure 5A,B, Appendix A). A similar situation occurs on L456F, which ranks second in terms of node enlargement and enhances the ACE2-binding affinity by −4.9 kJ/mol. The third ranked node enlargement occurs on N493Q. This residue change considerably enhances the binding affinity by −36.7 kJ/mol due to the formation of three HBs between RBD_Co__V2_:Q493 and ACE2:K31 and E35 (Figure 5C,E), which strengthens the electrostatic interactions by −49.9 kJ/mol (Appendix A). Y498Q enlarges the node size and enhances the ACE2-binding affinity (by −17.7 kJ/mol) because of the formation of multiple additional HBs with ACE2:D38, Q42, and K353, which strengthens the electrostatic interactions by −33.4 kJ/mol (Appendix A). P475A increases the node size due to the formation of a high-occupancy HB between RBD_Co__V2_:A475 and ACE2:S19 (Figure 5C), which strengthens the electrostatic interactions by −17.3 kJ/mol. Although D476G leads to a limited increase in the node size, it leads to an abnormal enhancement in the ACE2-binding affinity by −337.9 kJ/mol (Figure 4D). The reason for this is the loss of the long-range electrostatic repulsive interactions (Appendix A) with ACE2.

The cumulative number of contacts made by the non-conserved RBD interface residues is higher in RBD_Co__V2_-ACE2’s IRCN (63.7 and 60.2 with and without RBD_Co__V2_:G476, respectively) than in RBD_Co__V_-ACE2’s IRCN (41.3 and 39.1 with and without RBD_Co__V_:D476, respectively). For the RBD_Co__V_-ACE2 complex, the cumulative BFE values of the non-conserved RBD interface residues with and without RBD_Co__V_:D476 are 248.3 and −82.3 kJ/mol, respectively. For the RBD_Co__V2_-ACE2 complex, the corresponding values with and without RBD_Co__V2_:G476 are −165.9 and −158.6 kJ/mol, respectively. These results indicate that, although D476G plays an overwhelming role in enhancing the ACE2-binding affinity, there is a clear trend in which the increased close contacts on the non-conserved RBD_Co__V2_ interface residues considerably enhance the ACE2-binding affinity of RBD_CoV2_.

There are non-shared nodes, either from RBDs (light red) or from ACE2 (light green), between the two IRCNs (Figure 5A,B). Two non-shared nodes, RBD_CoV_:R439 and RBD_CoV2_:K417, are worth noting. More specifically, the direct contacts of RBD_Co__V_:R439 with ACE2:E329 are absent in the IRCN of RBD_Co__V2_-ACE2 due to R439N. This results in the loss of the direct HB and SB interactions and the indirect long-range electrostatic attractive interactions present in RBD_Co__V_-ACE2 (Figure 5C,D), thus reducing the ACE2-binding affinity of RBD_CoV2_ by 547.0 kJ/mol (Figure 4D and Appendix A). The residues at position 417 are RBD_CoV2_:K417 and RBD_CoV_:V417. Although both residues are located outside the RBM region, RBD_CoV2_:K417 makes close contacts with ACE2:D30 and H34 due to its longer, positively charged side chain. In particular, RBD_CoV2_:K417 forms two HBs and one SB with ACE2:D30 (Figure 5C,E), which together with the long-range electrostatic forces of attraction to ACE2, provide the most favorable contribution to the ACE2-binding affinity (−495.1 kJ/mol) among all the RBD_CoV__2_ residues (Figure 4B and Appendix A). In addition, it is worth noting that among all the residue changes, V417K makes the largest contribution to enhancing the ACE2-binding affinity of RBD_CoV__2_ (by −487.6 kJ/mol; see Figure 4D). Overall, the cumulative BFE values of the non-shared RBD residues in the IRCNs of RBD_CoV_-ACE2 and RBD_CoV2_-ACE2 are −560.6 and −504.1 kJ/mol (Appendix A), respectively, indicating that the non-shared RBD residues are more conducive to enhancing the ACE2-binding affinity of RBD_CoV_. Nevertheless, when taking all the RBD interface residues (including the conserved, non-conserved, and non-shared RBD nodes) into account, the cumulative BFE values are −431.9 and −779.2 kJ/mol, respectively (Appendix A). This indicates that the interface residues of RBD_CoV2_ are more conducive to enhancing the ACE2-binding affinity than those of RBD_CoV_. 

There are two (L45 and E329) and four (L79, D30, E35, and R393) non-shared ACE2 nodes (light green) in the IRCNs of RBD_CoV_-ACE2 (Figure 5A) and RBD_CoV2_-ACE2 (Figure 5B), respectively. Among them, the uncharged nodes (L45 and L79) only slightly enhance the RBD-binding affinity when compared with the corresponding ACE2 residues that make no close contact with the RBD (Appendix A). In contrast, the negatively/positively charged nodes greatly enhance/reduce the RBD-binding affinity when compared with the corresponding non-interface ACE2 residues (Appendix A). Despite more non-shared ACE2 nodes in the IRCN of RBD_CoV2_-ACE2, their cumulative BFE value (−119.6 kJ/mol) is less negative than that of the non-shared ACE2 nodes in the IRCN of RBD_CoV_-ACE2 (−164.4 kJ/mol). However, when taking all the ACE2 interface residues (non-shared and shared ACE2 nodes) into account, the cumulative BFE values are −308.5 and −246.8 kJ/mol, respectively, indicating that the ACE2 interface residues in RBD_CoV2_-ACE2 make an overall larger positive contribution to the RBD-binding affinity than those in RBD_CoV_-ACE2.

In summary, the RBD_CoV__2_-ACE2 complex has more close inter-atomic contacts across the binding interfaces and a tighter interface packing than RBD_CoV_-ACE2. When compared with RBD_CoV_-ACE2, RBD_CoV2_-ACE2 has more favorable interface vdW interactions (Appendix A). Nevertheless, the enhanced binding affinity of RBD_CoV2_-ACE2 is primarily determined by the significantly strengthened electrostatic attractive interactions occurring on the interface residues (Appendix A). Several changes play crucial roles in strengthening the electrostatic interactions between the RBD_CoV2_ interface and ACE2: the gain of HBs and SB and of the long-range electrostatic attractive forces due to V417K, the loss of the electrostatic repulsive forces due to D476G, the gain of HBs due to N493Q and Y498Q, and the increased number of HBs formed by the ACE2 interface residues with RBD_CoV2_.

## 4. Discussion

Although the experimentally determined crystal structures of RBD_Co__V_-ACE2 and RBD_Co__V2_-ACE2 [13,14] show overall similar conformations and nearly identical binding modes, our MD simulation-based analyses reveal their distinctly different dynamic and thermodynamic behaviors. When compared to RBD_Co__V2_-ACE2, RBD_Co__V_-ACE2 features enhanced global structural fluctuations and has larger conformational entropy and conformational diversity. During MD simulations the individual binding partners also underwent larger structural fluctuations in RBD_CoV_-ACE2 than in RBD_Co__V2_-ACE2, and in particular, the two binding partners experienced larger relative positional movements in the former complex. This suggests that the enhanced protein–protein association could enhance the structural stability of the individual binding partners and the entire complex. Undoubtedly, it is the amino-acid-sequence differences between RBD_Co__V_ and RBD_Co__V2_ and the resulting changes in RBD physicochemical properties and in RBD-ACE2 interaction strengths that give rise to the observed different dynamic and thermodynamic behaviors of the two complexes, which in turn may also have impacts on interactions and binding affinities between RBDs and ACE2.

The detailed comparison of the calculated values of various MM-PBSA energy terms (Table 1) reveals that (i) the inter-protein electrostatic interactions determine the high-affinity bindings of both RBDs to ACE2 and, (ii) the significantly strengthened electrostatic attractive interactions between RBD_CoV__2_ and ACE2 determine the higher affinity of RBD_CoV__2_-ACE2 compared with RBD_Co__V_-ACE2. ACE2 is heavily negatively charged at pH 7.4 with the predicted net charges of −25 and −24 in RBD_CoV_-ACE2 and RBD_Co__V2_-ACE2, respectively, thus generating an overall extremely intense negative electrostatic potential around itself (Appendix A). Both RBD_CoV_ and RBD_CoV__2_ have a net charge of +2, thus generating an overall moderately intense positive electrostatic potential while showing different distributions of localized positive and negative electrostatic potentials (Appendix A). Consequently, ACE2 has strong electrostatic forces of attraction to both RBDs. The observed difference in the inter-protein electrostatic interaction strengths between the two complexes could be attributed to the different distributions of the positive and negative electrostatic potentials on the two RBDs. Furthermore, the electrostatic potential distributions depend on the structural locations of the positively and negatively charged residues. Therefore, it is reasonable to contemplate a scenario in which the intense negative electrostatic potential around ACE2 could accelerate the diffusion of both net positively charged RBDs toward ACE2 through a strong electrostatic attraction. This would facilitate the initial recognition and subsequent orientation adjustment between them. The different electrostatic interaction strengths resulting from different distributions of the charged residues on RBD_Co__V_ and RBD_CoV__2_ determine the final difference in ACE2-binding affinities of the two RBDs. 

A previous study [62] on the electrostatic features of the two complexes reveals that the differences in distributions of the charged residues and their electric field line density between RBD_CoV_ and RBD_CoV__2_ determine the stronger electrostatic attractive forces of ACE2 to RBD_CoV2_, thus supporting our speculation. In the current study, we found that, although the charged residues commonly exhibit large BFE values, there is a trend that the magnitudes of the absolute values of BFE largely depend on the distance of the charged residues to the binding interfaces (Figure 4A,B). In fact, it is the distance-dependent changes in the electrostatic interaction strength of the charged residues that give rise to the different degrees of contribution to the binding affinity (Appendix A). Although with few exceptions, a general trend is observable that the closer the distance of the negatively charged ACE2 residues and positively charged RBD residues to the binding interfaces, the stronger the electrostatic attractive interactions with the other partners (Appendix A), and hence, the greater the positive contribution to the affinity (Appendix A). A similar trend also occurs for the positively charged ACE2 residues and negatively charged RBD residues, in which the electrostatic repulsive interactions increase as the distance to the interfaces decreases. 

A comparison between the IRCNs of the two complexes reveals that the RBD residue changes result in more intensive interface contacts and a tighter interface packing in RBD_CoV2_-ACE2 than in RBD_CoV_-ACE2. On the one hand, these interface changes explain the reduced inter-protein positional movements in RBD_CoV2_-ACE2, and on the other hand, they significantly enhance the overall electrostatic interaction strength of the interface residues with the other partners. These observations are consistent with a previous simulation study showing that RBD_CoV2_-ACE2 has greater electrostatic complementarity and enhanced hydrophobic packing at the interfaces than the RBD_CoV_-ACE2 complex [27]. Interestingly, a crystal structure of the chimeric RBD (with the core from RBD_CoV_ and RBM from RBD_CoV2_) in complex with human ACE2 also reveals a tighter ACE2 binding by the chimeric RBD than by RBD_CoV_ [28]. 

For RBD_CoV2_-ACE2 and RBD_CoV_-ACE2, the values of the inter-protein electrostatic interaction energy are −2597.0 and −2338.4 kJ/mol (Table 1), respectively. The cumulative values of the electrostatic interaction energy of all the IRCN-forming residues (i.e., the interface residues) are −1258.6 and −799.8 kJ/mol (Appendix A), respectively, and those of all the non-interface residues are −1338.4 and −1538.7 kJ/mol, respectively. Therefore, despite the importance of the long-range indirect electrostatic attractive interactions in promoting the high-affinity bindings of both RBDs to ACE2, the stronger electrostatic attractive forces occurring on the interface residues of RBD_CoV2_-ACE2 dictate the overall stronger inter-protein electrostatic interactions of RBD_CoV2_-ACE2 compared with RBD_CoV_-ACE2. Further comprehensive comparative analyses of IRCNs, IRCN-related interactions, and energy components of IRCN-forming residues between the two complexes reveal that the difference in the electrostatic interaction strength depends on the charge properties of the interface residues, the number of close contacts on the charged residues, and whether or not the close contacts involve the formation of HB and SB.

Of interest is that for the RBM residues (438-506) of the RBD_CoV2_ and RBD_Co__V_, the cumulative BFE values are −574.3 and −1313.7 kJ/mol, respectively, and the cumulative values of the electrostatic interaction energy are −538.7 and −1300.8 kJ/mol (Appendix A), respectively. Therefore, the stronger electrostatic interactions of the RBD_CoV_ RBM with ACE2 determine its larger positive contribution to the ACE2-binding affinity. The net charges of RBMs in RBD_CoV_ and RBD_CoV2_ are +3 and +1, respectively, thus explaining why RBM has stronger electrostatic forces of attraction to ACE2 in RBD_Co__V_-ACE2. Why does the RBD_Co__V2_ interface have stronger electrostatic forces of attraction to ACE2 than the RBD_Co__V_ interface (−812.2 vs. −451.9 kJ/mol; see Appendix A)? The reasons for this are as follows. The first is that the RBD interface residues (i.e., IRCN-forming residues from RBDs) identified in this study include only a fraction of the RBM residues and most of the RBM residues make no direct contact with ACE2 (Figure 1D). The second is that the net charges of the binding interfaces of RBD_CoV_ and RBD_CoV2_ are 0 (determined by R439 and D476; see Figure 5A) and +1 (determined by K417 that does not belong to RBM), respectively. The third is that RBD_CoV2_ interface residues form more HBs with ACE2 than RBD_CoV_ interface residues (Figure 5C–E). We could thus conclude that it is the effective RBD interface rather than RBM that dominates the higher ACE2-binding affinity of RBD_CoV__2_ than of RBD_CoV_. This suggests that when estimating the changes in the ACE2-binding affinity between different RBDs or RBD mutants, one should first pay attention to the changes in the electrostatic interactions caused by the residue changes/mutations at the RBD interface, then those near the interface, and finally those located distally from the interface, rather than simply focusing on the residue changes/mutations in RBM.

Although we do not calculate the BFEs between human ACE2 and RBDs of various variants of concern (VOC) of SARS-CoV-2, our findings can still facilitate the explanation of the experimentally observed changes in ACE2-binding affinities of different VOC RBDs from the perspective of the electrostatic interaction change principle. For example, RBD of the Delta variant (B.1.617.2 lineage) contains two residue mutations, L452R and T478K [63], which result in the gain of two positively charged residues near the binding interfaces, and hence, could greatly strengthen the electrostatic forces of attraction to ACE2. Of interest is that in RBD_Co__V_ the corresponding residues are positively charged K452 and K478, which, compared with RBD_Co__V2_:L452 and T478, enhance the ACE2-binding affinity of RBD_Co__V_ (by −408.8 and −306.3 kJ/mol, respectively; see Figure 4D) through significantly strengthening their electrostatic forces of attraction to ACE2 (by −413.3 and −301.9 kJ/mol, respectively; see Appendix A). These observations may help explain the experimentally observed 1.2-4.6-fold increase in the ACE2 affinity of the Delta RBD compared to the WT RBD_CoV2_ [64,65,66].

The recently emerged VOC Omicron (B.1.1.529 lineage) contains 37 residue mutations in the spike protein, of which 15 are in the RBD and 11 are near/at the binding interfaces [67,68]. Of the four mutations (G339D, S371L, S373P, S375F) with locations far from the binding interfaces, G339D would largely reduce the ACE2 affinity due to the long-range electrostatic repulsion of D339 with ACE2 (as observed for E340 in both RBD_Co__V_ and RBD_Co__V2_; see Figure 4B), whereas the effects of the other three mutations could be ignored. Of the 11 mutations (K417N, N440K, G446S, S477N, T478K, E484A, Q493R, G496S, Q498R, N501Y, and Y505H) located near/at the RBD interface, six (K417N, N440K, T478K, E484A, Q493R, and Q498R) involve charge changes and could greatly impact the ACE2 affinity. Specifically, E484A would overcompensate for the negative effect of G339D because the distance of residue 484 to the binding interfaces is shorter than that of residue 339 (Figure 4B). Thus, E484A could lead to a greater loss of the electrostatic repulsion with ACE2 compared with the added electrostatic repulsion by G339D. Although all the other five mutations involve either the gain or loss of the positively charged residues, they lead to a net gain of three positive charges, thus increasing the number of net positive charges from +2 in the WT RBD_Co__V2_ to +5 in the Omicron RBD. This, in conjunction with the close distances/contacts of all the newly acquired positively charged residues to/with ACE2, could greatly enhance the ACE2-binding affinity of the Omicron RBD. The above inference is confirmed by the experimental measurements showing that the Omicron RBD enhances the ACE2 affinity by 1.4–2.4 folds compared to the WT RBD_CoV2_ [64,65,66]. In addition, a computational study [69] shows that (i) the mutations involving the gain of the positively charged residues (N440K, T478K, Q493R, and Q498R) and loss of the negatively charged residue (E484A) collectively enhance, although to different extents, the ACE2-binding affinity, and (ii) the gain of the negatively charged residue (G339D) reduces the ACE2 affinity to an extent that can be over-compensated for with the increased affinity contributed by E484A, in line with our electrostatic interaction change principle-based inference. In addition, it is worth noting that a Cryo-EM study reveals [65] that Q493R and Q498R lead to the formation of new HBs and SBs with ACE2:E35 and D38, respectively. This can over offset the local electrostatic repulsion with ACE2:K31 and K353, respectively, and ultimately results in a large increase in the electrostatic interaction strength with ACE2 and enhances the affinity.

## 5. Conclusions

Through comprehensive comparative analysis of our computational results, we conclude that it is the RBD residue changes at the binding interface that lead to the overall stronger electrostatic force of attraction of ACE2 to RBD_Co__V2_ than to RBD_Co__V_. The strengthened electrostatic force, on the one hand tightens the interface packing and reduces the structural fluctuations of RBD_CoV2_-ACE2, and on the other hand, enhances the ACE2-binding affinity of RBD_CoV2_. Although the RBD residue changes involved in the charge changes can significantly impact the inter-protein electrostatic interaction strength, and hence, the binding affinity, the extent of the impact largely depends on the distance of the residue to the binding interfaces (i.e., the extent of the impact increases/decreases as the distance decreases/increases). Furthermore, the formation or destruction of the interface HBs and SBs caused by RBD residue changes can largely impact the inter-protein electrostatic interactions. Our findings not only shed light on the mechanical and energetic mechanisms responsible for modulating the inter-protein interaction strengths and binding affinities of the two RBD-ACE2 complexes, but they can also help predict the binding affinity changes of different VOC RBDs to ACE2.

## Figures and Tables

**Figure 1 cells-11-01274-f001:**
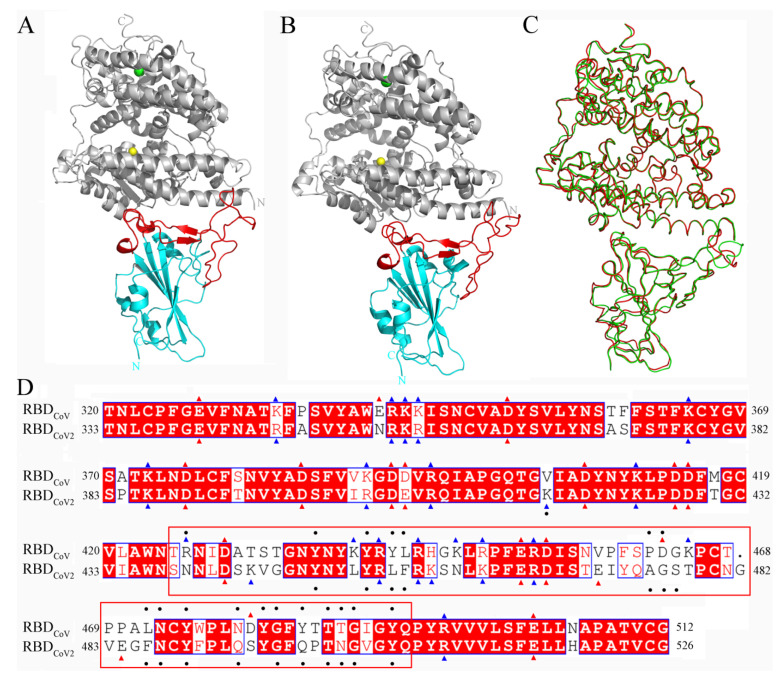
Structures of the RBD_CoV_-ACE2 and RBD_CoV__2_-ACE2 complexes and sequence alignment of RBD_CoV_ and RBD_CoV__2_. (**A**,**B**) Cartoon representations of the complete complex structures of RBD_CoV_-ACE2 (modeled based on the crystal structure with PDB ID 2AJF [13]) and RBD_CoV2_-ACE2 (PDB ID: 6M0J [14]), respectively. ACE2 is colored gray, with Zn^2+^ and Cl^−^ ions represented as spheres in yellow and green, respectively; cores and RBMs of both RBDs are colored cyan and red, respectively. (**C**) Backbone superposition of RBD_CoV_-ACE2 (red) and RBD_CoV2_-ACE2 (green). (**D**) Structure-based sequence alignment of RBD_CoV_ and RBD_CoV2_. The identical residues are white on a red background and the similar residues are red on a white background; the negatively and positively charged residues are indicated by red and blue triangles, respectively. The ACE2-contacting residues (or RBD interface residues) identified in this work are indicated by black dots; RBM (residues 438–506 according to residue numbering of RBD_CoV2_) is highlighted by enclosure with a red box.

**Figure 2 cells-11-01274-f002:**
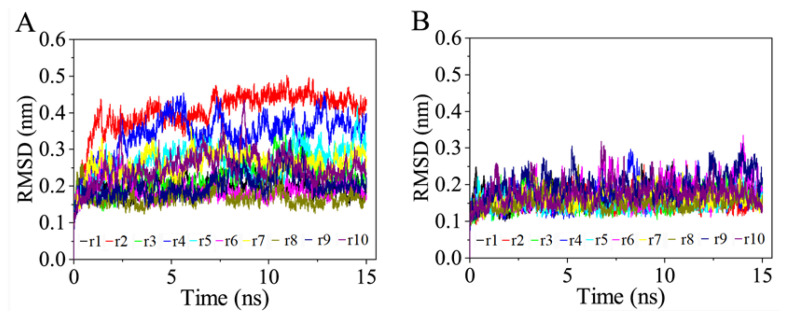
Time evolution of the C_α_ root mean square deviation (RMSD) values of the two complexes relative to their respective starting structures calculated from the 10 independent MD simulation replicas (r1–10). (**A**) RBD_CoV_-ACE2 complex. (**B**) RBD_CoV2_-ACE2 complex.

**Figure 3 cells-11-01274-f003:**
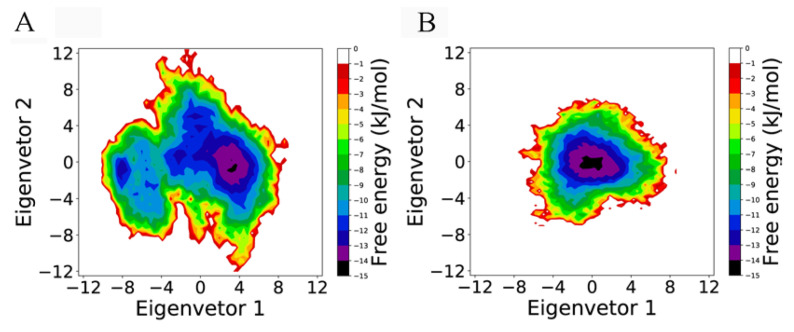
Free energy landscapes (FELs) of the two complexes as a function of the projection of the single joined equilibrium trajectory onto the essential subspace spanned by eigenvectors 1 and 2. (**A**) FEL of the RBD_CoV_-ACE2 complex. (**B**) FEL of the RBD_CoV2_-ACE2 complex. The color bar represents the relative free energy value in kJ/mol.

**Figure 4 cells-11-01274-f004:**
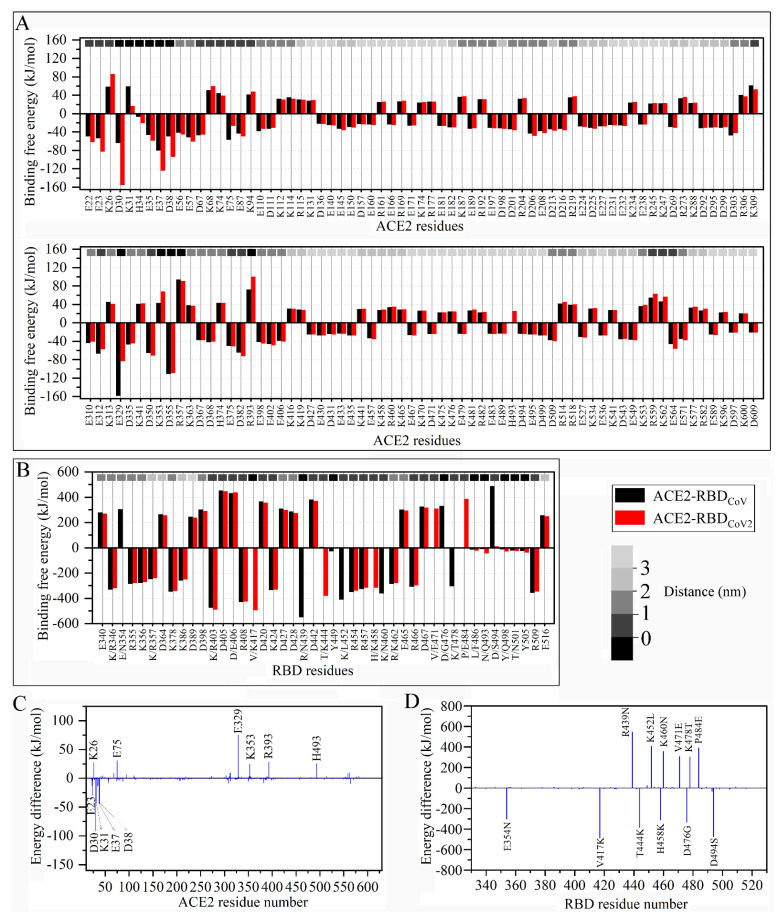
Residue binding free energy (BFE) average values and their differences between the RBD_Co__V_-ACE2 and RBD_Co__V2_-ACE2 complexes. (**A**) Residue BFE average values of the RBD_CoV_-bound (black) and RBD_CoV2_-bound (red) ACE2s. (**B**) Residue BFE average values of RBD_CoV_ (black) and RBD_CoV2_ (red). In (**A**,**B**), only the residues with BFE values greater than 20 kJ/mol or lower than −20 kJ/mol are shown. The distances of residues to the binding interfaces are marked by small rectangles of different shades of gray along the top horizontal axis, with the black rectangles representing the distance of 0 nm (corresponding to the interface residues identified in this work) and those of reduced gray representing the increased distance to the binding interfaces as indicated by the gray bar. Residue labels in (**B**) are shown as single-letter amino acid codes along with the residue number according to RBD_CoV2_ numbering if the two residues are identical at the structurally equivalent positions of the two complexes (see Figure 1D), and if different ones are shown as RBD_CoV_ residue/RBD_CoV2_ residue, plus the residue number of RBD_CoV2_. (**C**) Per-residue BFE difference calculated by subtracting the value of the residue in the RBD_Co__V_-bound ACE2 from that in the RBD_Co__V2_-bound ACE2. (**D**) Per-residue, the BFE difference is calculated by subtracting the value of the residue in the ACE2-bound RBD_Co__V_ from that of the structurally equivalent residue in the ACE2-bound RBD_Co__V2_. Residue changes that result in a significant energy difference are labeled as “mutation” representation (e.g., E354N from K354 in RBD_CoV_ to N354 in RBD_CoV2_) but without the implication of residue mutation.

**Figure 5 cells-11-01274-f005:**
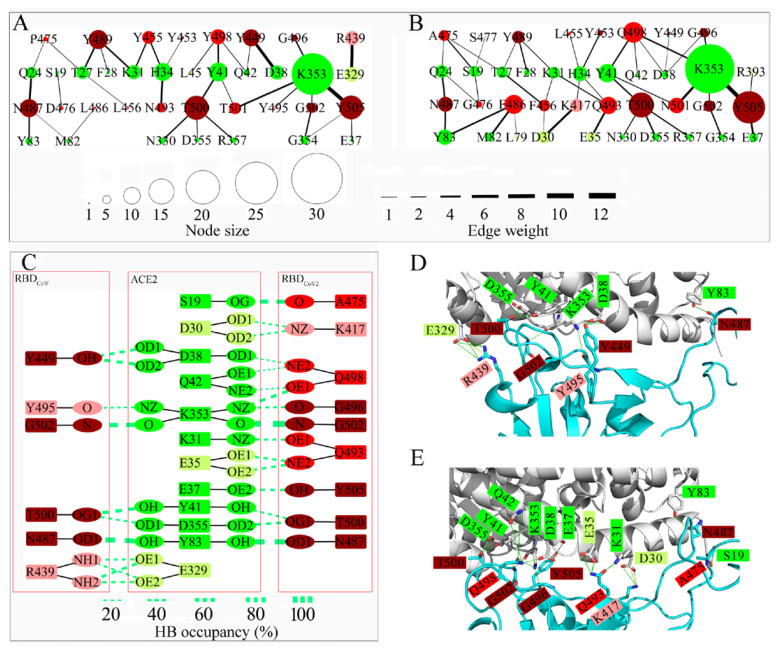
Interface residue contact networks (IRCNs) and special interactions across the binding interfaces of the RBD_CoV_-ACE2 and RBD_Co__V2_-ACE2 complexes. (**A**,**B**) IRCNs constructed over the 100 representative structures of the RBD_CoV_-ACE2 and RBD_CoV2_-ACE2 complexes, respectively. Nodes are colored as follows: shared ACE2 residues between the two IRCNs: green; non-shared ACE2 residues: light green; conserved RBD residues between the two IRCNs: dark red; non-conserved/changed RBD residue: red; non-shared RBD residues: light red. The node size represents the total number of close inter-atomic contacts occurring on a residue/node. The edge weight represents the average number of close inter-atomic contacts between the connected residues/nodes. (**C**) Interface hydrogen bonds (HBs) with an occupancy greater than 20%, which were identified over the respective 100 representative structures of the two complexes. HB–forming residues are colored the same as in (**A**,**B**). HBs are represented by green dash lines connecting between the donor and/or acceptor atoms (shown as atom names in PDB format), with the line thickness representing the degree of the HB occupancy. (**D**,**E**) Structural locations of the interface HBs and salt bridges (SBs) in RBD_Co__V_-ACE2 and RBD_Co__V2_-ACE2, respectively. The representative structures of the two complexes are shown as cartoon representations, with ACE2 and RBDs colored gray and cyan, respectively. HB-forming and SB-forming residues are rendered as stick models, with oxygen and nitrogen atoms colored red and blue, respectively. HBs and SBs are represented by green and yellow dashed lines, respectively. Residue labels are colored the same as in (**A**,**B**).

**Table 1 cells-11-01274-t001:** Average values and corresponding standard deviations (shown in parentheses) of various MM-PBSA energy terms (kJ/mol) for the RBD_Co__V_-ACE2 and RBD_Co__V2_-ACE2 complexes calculated over the respective 100 representative structures.

Energy Term ^a^	RBD_CoV_-ACE2	RBD_CoV2_-ACE2
Δ*E*_vdW_	−360.2 (27.8)	−376.0 (24.0)
Δ*E*_elec_	−2338.4 (146.8)	−2597.0 (146.7)
Δ*G*_polar_	452.3 (85.5)	561.9 (84.8)
Δ*G*_non-polar_	−42.9 (4.2)	−43.9 (3.4)
Δ*G*_binding_	−2289.2 (129.2)	−2455.0 (103.2)

^a^ For detailed explanations of the energy terms, see Equation (2) in Section 2.4.

## Data Availability

All data are contained within the article or its Appendix A as Figures or Tables.

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
