# Peer review of "Mechanistic Origin of Different Binding Affinities of SARS-CoV and SARS-CoV-2 Spike RBDs to Human ACE2"

_cells, 2022, doi:10.3390/cells11081274_

Round 1

Reviewer 1 Report

In this article titled “Mechanistic origin of different binding affinities of SARS-CoV 2 and SARS-CoV-2 spike RBDs to human ACE2,” the authors the structural and energetic basis of the differences in the affinity of the RBD-ACE2 complex of SARS-CoV and SARS-CoV-2. They carried out MD simulations with the MM-PBSA method and interface residue contact network analysis. The results showed that the attractive electrostatic interactions are the dominant force in both complexes, and this contribution is also responsible for the higher affinity of this complex in SARS-CoV-2. In addition, they found the structural features that possibly explain the higher relationship of the RBD-ACE2 complex of SARS-CoV2. The research is interesting since it tries to explain the differences in affinity of the RBD-ACE2 complex of SARS-CoV and SARS-CoV-2 using theoretical methods. Although the paper is well written, it needs more work to be acceptable.

Major concerns:

1). In the introduction, add and discuss recent literature related to the RBD-ACE2 complexes of SARS-CoV-2; J Mol Graph Model. 2021 Sep; 107: 107970; Sci Rep 2021 25;11(1):4659

2) In material and method, describe based on what criterion PDB entries 2AJF and 6M0J.

3) In methods explain why a NaCl concentration of 0.15 M was selected.

4) In the methods mentioned, describe why AMBER99SB-ILDN force field was employed and the advantages of the other forcefields present in GROMACS 5.1.4.

5) In methods provide more details about the concatenation of the multiple trajectories only considering the equilibrated portion. Mention the gromacs command (maybe gmx trjcat) employed for this and why the alfa-carbon and no backbone atoms were selected.

6) In methods, describe how the entropic component was determined? and if this contribution was really included in the determination of the binding free energy

7) Mention the value for the implicit solvent (PB) model employed

7) Mention in the manuscript whether the binding free energy values are absolute or relative binding free energies.

8) Explain how the representative conformations were obtained for all the complexes? Did the authors perform clustering analysis? If yes, this was achieved over the concatenated trajectories or for each trajectory?

9) Discuss how the contacts map for both complexes before and after MD simulations change?

10) Add a graph showing how the energies reached equilibrium.

Author Response

Response to Reviewer 1 Comments

Reviewer Report 1

Comments and Suggestions for Authors

In this article titled “Mechanistic origin of different binding affinities of SARS-CoV 2 and SARS-CoV-2 spike RBDs to human ACE2,” the authors the structural and energetic basis of the differences in the affinity of the RBD-ACE2 complex of SARS-CoV and SARS-CoV-2. They carried out MD simulations with the MM-PBSA method and interface residue contact network analysis. The results showed that the attractive electrostatic interactions are the dominant force in both complexes, and this contribution is also responsible for the higher affinity of this complex in SARS-CoV-2. In addition, they found the structural features that possibly explain the higher relationship of the RBD-ACE2 complex of SARS-CoV2. The research is interesting since it tries to explain the differences in affinity of the RBD-ACE2 complex of SARS-CoV and SARS-CoV-2 using theoretical methods. Although the paper is well written, it needs more work to be acceptable.

Major concerns:

1). In the introduction, add and discuss recent literature related to the RBD-ACE2 complexes of SARS-CoV-2; J Mol Graph Model. 2021 Sep; 107: 107970; Sci Rep 2021 25;11(1):4659

Response:

The above mentioned works have been introduced in the introduction section.

2) In material and method, describe based on what criterion PDB entries 2AJF and 6M0J.

Response:

We have explained the reasons for taking PDB entries 2AJF and 6M0J as the research objects: “which were chosen because of their high resolutions (2.90 and 2.45 Å, respectively) and the wild-type binding partners in both complex structures.”

3) In methods explain why a NaCl concentration of 0.15 M was selected.

Response:

We have rephrased the sentence “The net charges of both systems were neutralized with NaCl at the physiological concentration of 0.15 M” to “The net charges of both systems were neutralized with a 0.15 M concentration of NaCl to mimic the physiological conditions” to explain why a NaCl concentration of 0.15 M was selected.

4) In the methods mentioned, describe why AMBER99SB-ILDN force field was employed and the advantages of the other forcefields present in GROMACS 5.1.4.

Response:

The AMBER99SB-ILDN force field was employed because its performance is likely to be among the best of currently available empirical all-atom force fields for protein MD simulations due to the improvement in side-chain torsion potentials of amino acids. The explanation and relevant references have been added to the Materials and Methods section.

5) In methods provide more details about the concatenation of the multiple trajectories only considering the equilibrated portion. Mention the gromacs command (maybe gmx trjcat) employed for this and why the alfa-carbon and no backbone atoms were selected.

Response:

The GROMACS command ‘trjcat’ was mentioned. The explanation of why the Ca rather than backbone atoms were selected for PCA analysis was added to section 2.3, i.e., “For each complex system, the equilibrated trajectory portion of each of the 10 replicas was concatenated into a single joined equilibrium trajectory using the GROMACS tool ‘gmx trjcat’. The principal component analysis (PCA) was performed on the Ca atoms of the single joined equilibrium trajectory using the GROMACS tools ‘gmx covar’ and ‘gmx anaeig’; the selection of the Ca rather than backbone atoms is sufficient to provide reliable information about the large concerted protein motions during simulations while reducing the computation cost.”.

6) In methods, describe how the entropic component was determined? and if this contribution was really included in the determination of the binding free energy

Response:

The change in the solute entropy upon binding is often approximated with the normal mode method using a few selected snapshots, and such contribution was not included in our binding free energy calculations. In the section 2.3, we have explained why the entropy contribution was omitted.

7) Mention the value for the implicit solvent (PB) model employed

Response:

The parameters for calculating Gpolar using PB model have been added to the section 2.4.

7) Mention in the manuscript whether the binding free energy values are absolute or relative binding free energies.

Response:

We have pointed clearly that the calculated binding free energy values in our work are the relative binding free energies (due to the lack of the entropy estimate) as suggested by the reviewer, i.e., “As a result, the BFE values calculated in this work are the ones of the relative binding energies, which, although unphysical, can be used to compare binding affinities of different ligands to a common receptor.”

8) Explain how the representative conformations were obtained for all the complexes? Did the authors perform clustering analysis? If yes, this was achieved over the concatenated trajectories or for each trajectory?

Response:

We are sorry for the unclear expression of obtaining the representative conformations. In this work we did not perform clustering analysis but obtained the representative conformations through randomly extracting 100 structures from the global free energy minimum in the FEL constructed from the single joined equilibrium trajectory of each complex. For details, see the last paragraphs of the section 2.3 and section 3.2.

9) Discuss how the contacts map for both complexes before and after MD simulations change?

Response:

We did not describe and discuss the differences in the interface residue contact networks (IRCN) before and after MD simulations because i) the interface residue contacts in the crystal structures of the two RBD-ACE2 complexes (before MD simulation) have been described in the literature (references 13 and 14 in the revised manuscript) by authors who determined the complex structures and, ii) the IRCNs identified based on the crystal complex structures can only reflect close residue-residue contacts in the crystallization conditions, whereas IRCNs constructed over the representative conformations sampled by MD simulations in the solvent conditions reflect more physiological realistic interface residue interactions. Therefore, we focus only on the IRNCs constructed from the representative conformations sampled by MD simulations that mimic the physiological reality.

10) Add a graph showing how the energies reached equilibrium.

Response:

As we know, Cα or backbone RMSD relative to the starting structure is one of the most frequently used indicator for assessing the convergence of a simulation, although the time-dependent changes in the other structural and energy parameters during a simulation can also be used. Since the main aim of our MD simulation is to obtain the representative conformations for the binding free energy calculation and IRCN construction, we believe that equilibrated trajectories obtained based on the Cα RMSD values can reflect the equilibrium distribution of the conformations sampled by MD simulations on the time scale. Therefore, it is not necessary to assess the convergence of simulations based on energy parameters.

Reviewer 2 Report

Importance of Research/Value to the Field: 3 = strong

Importance of Research/Value to the Field: 2 = average

Originality: 1 = need improvements  

Technical Rigor: 2 = Average

Proper Citation of Previous Work: 2 = Average

Clarity of Writing/English Usage: 3 = strong

Quality of Figures: 2 = Average

This manuscript utilised molecular dynamics simulation to study the difference between the SARS-COV and SARS-CoV-2 in terms of binding affinity with ACE2. The manuscript addressed an important question that is related to the current pandemic,  the result showed a clear difference between the two viruses. However, a major problem in this manuscript is the novelty, as there are numerous articles already reported the difference in terms of binding affinity and interface residue contact network (Amin et al., 2020; Yan et al., 2021; Sinha et al., 2022). Therefore, novelty is an issue in this manuscript.

Furthermore, the unphysical binding free energy for point residue and the system is a major concern. As mentioned below, much literature reported significantly lower binding free energy.

The lineage of SARS-CoV and SARS-CoV-2 do not support that SARS-CoV-2 evolved from SARS-CoV. Therefore, there is no residue substitution in this study but this terminology was used throughout the manuscript.

Major:

Numerous simulation and experimentsal reports showed that the binding energy of SARS-CoV-2 RBD and ACE2 is less than -500 kJ mol‑1 (wang et al., 2020; Ali et al., 2020; Zhang et al., 2021; Verma et al., 2021, Sinha et al., 2022). The free binding energy reported in this report is unphysical and the authors should address why their binding energy is much higher than the range reported in previous studies.  “The absolute energy value of the charged residues (in a range of about 240-550 kJ/mol-1)” is suggesting the residue is forming a covalent bond with the ACE2 residues. Many RBD-ACE2 point mutation studies showed the value is around <10 kJ mol-1.

Figure 4B – The RBD residue is misrepresented as this study is not a mutation study. For example, the convention “T444K” meant T substitute with K at position 444. The meaning of this graph is the structure-based alignment of SARS-CoV and SARS-CoV-2, where the similar alignment position had different amino acid residues, but by no means this is a substitution. Furthermore, the author did not mention which position alignment they used.

The discussion between line 704 to line 710: The authors only used binding free energy to conclude the results. However, physical evidence such as the structural change and the binding angle between RBD and ACE2 will enhance their conclusion.

Minor:

Keywords: “SARS-CoV-2 variants” – there is no variants in the study. “amino acid residue substitutions “ – this is a wild-type SARS-CoV and SARS-CoV-2 study, there are no amino acid residue substitutions

Line 304 – please clarified “single joined equilibrium trajectory of each complex”, does this mean only a single model was used to calculate FEL?

Figure 4C – the bottom x-axis had different font.

Figure 5 – The node size and the hydrogen line should have a scale. Furthermore, figure 5C, the text is not aligned, what does OG, OD1, OD2 mean?

Line 478 – Hydrogen bond is a dipole-dipole interaction, therefore should not use the term of donor or acceptor as there is no transfer of proton or electron.

Line 767 – This is not a residue substitution and the study did not give evidence of conformation change.

Author Response

Response to Reviewer 2 Comments

Reviewer Report 2

This manuscript utilised molecular dynamics simulation to study the difference between the SARS-COV and SARS-CoV-2 in terms of binding affinity with ACE2. The manuscript addressed an important question that is related to the current pandemic,  the result showed a clear difference between the two viruses. However, a major problem in this manuscript is the novelty, as there are numerous articles already reported the difference in terms of binding affinity and interface residue contact network (Amin et al., 2020; Yan et al., 2021; Sinha et al., 2022). Therefore, novelty is an issue in this manuscript.

Response:

Thanks for pointing out the weakness of our manuscript. We admit that our results are not innovative enough in terms of binding affinity and interface residue interactions. However, after extensively and carefully reading the available literature, including the above papers mentioned by the reviewer, we found that most of the studies focus on investigating the difference in the binding affinity of the wild-type SARS-CoV-2 and its variants to human ACE2 while only a few studies focus on addressing the mechanistic reasons for different binding affinities of the two RBDs of SARS-CoV and SARS-CoV-2 to the same receptor ACE2, although the electrostatic interactions, including the interface hydrogens and salt bridges were commonly found to have a positive effect on enhancing affinity. Our work aims to explore the mechanistic origin of different affinities of SARS-CoV and SARS-CoV-2 spike RBDs to human ACE2 through comprehensive comparative analyses in terms of the dynamics, thermodynamics, energetics, and interface residue contact networks between the two complex systems. Our results complement previous studies, and more importantly shed light on the dynamic and energetic aspects of the modulation mechanism of RBD-ACE2 interactions and explain why SARS-CoV-2 RBD enhances its ACE2-binding affinity to ACE2 in comparison with SARS-CoV RBD.

Furthermore, the unphysical binding free energy for point residue and the system is a major concern. As mentioned below, much literature reported significantly lower binding free energy.

Response:

The magnitudes of the calculated binding free energy values of the point residue and the system merely represent the relative levels of binding energies; however, we believe that the calculated relative binding free energy values for the two complex systems can be utilized to explain the difference in the binding affinity due to the same computational protocol applied to the two systems.

The lineage of SARS-CoV and SARS-CoV-2 do not support that SARS-CoV-2 evolved from SARS-CoV. Therefore, there is no residue substitution in this study but this terminology was used throughout the manuscript.

Response:

Thank you for pointing this out. We have replaced the residue substitution with residue change in the revised manuscript.

Major:

Numerous simulation and experimentsal reports showed that the binding energy of SARS-CoV-2 RBD and ACE2 is less than -500 kJ mol‑1 (wang et al., 2020; Ali et al., 2020; Zhang et al., 2021; Verma et al., 2021, Sinha et al., 2022). The free binding energy reported in this report is unphysical and the authors should address why their binding energy is much higher than the range reported in previous studies.  “The absolute energy value of the charged residues (in a range of about 240-550 kJ/mol-1)” is suggesting the residue is forming a covalent bond with the ACE2 residues. Many RBD-ACE2 point mutation studies showed the value is around <10 kJ mol-1.

Response:

Indeed, the calculated binding free energy values of SARS-CoV-2 RBD and ACE2 in the above mentioned papers ranges from tens to hundreds of kJ/mol, with the magnitude of energy values being one to two orders of magnitude smaller than ours. The different magnitudes of the calculated energy values are most likely caused by different input parameters for MM-PB/GBSA calculations, in particular the values of the solute dielectric constant in the vacuum electrostatic energy calculation and the solute dielectric constant in the polar solvation free energy calculation. Unfortunately, most of the previous reports do not provide the detailed input parameters for MM-PB/GBSA calculations. In the study by Zhang et al., (Briefings in Bioinformatics, 22(6), 2021, 1–13), the values (shown in Table 1 in this paper) for energy terms of â–³EvdW (-382.6 kJ/mol), â–³Eeel (-2571.7 kJ/mol), and â–³Gnp (-42.2 kJ/mol) are very similar in magnitude to those of the respective terms (â–³EvdW, â–³Eelec, and â–³Gnon-polar of -376.0, -2597.0, and -43.9 kJ/mol, respectively) calculated in our work; however, the values of polar solvation free energy â–³Gpolar/p are very different between their (2791.4 kJ/mol) and our work (561.9 kJ/mol), which result in the significantly lower magnitude of the total binding free energy in their work (-68.6 kJ/mol) than in ours (-2455.0 kJ/mol). Since the paper by Zhang et al. only mentioned “The internal and external dielectric constants were set to 1 and 80, respectively, for both PBSA and GBSA calculations”, the values of the solute dielectric constant in the vacuum electrostatic energy calculation and of the solvent dielectric constant in the polar solvation free energy calculation are the same as those used in our work, i.e., 1 and 80, respectively; however, we assigned the solute dielectric constant value of 4 in the polar solvation free energy calculation, which results in a significantly lower value of the polar solvation free energy term than that in the study by Zhang et al. presumably using the solute dielectric constant value of 1. Anyway, the use of different solute dielectric constants for computing the polar solvation free energy term does not impair the comparison between the binding affinities of different ligands to a common receptor when the same computational protocol (or input parameters) is used for all systems due to the calculated total binding free energy values are the relative ones that reflect the relative differences between binding affinities (Kumari et al., J. Chem. Inf. Model. 2014, 54, 1951−1962).

  We are sorry for the misleading description “The absolute energy value of the charged residues (in a range of about 240-550 kJ/mol-1)”, in which “absolute energy value” does not mean the absolute binding free energy while represent the absolute value of the calculated residue binding free energy (DGresidue) without regard to its sign, i.e., |DGresidue| that reflects the magnitude of the relative contribution of a residue to the total binding affinity. The values of residue binding free energies are also unphysical while can be used to distinguish differences in the degree of contribution to the total binding free energy among different residues.

  We have pointed clearly out that the calculated binding free energy is the relative one and replaced “absolute energy value” with “absolute value of the binding free energy” in the revised manuscript.

Figure 4B – The RBD residue is misrepresented as this study is not a mutation study. For example, the convention “T444K” meant T substitute with K at position 444. The meaning of this graph is the structure-based alignment of SARS-CoV and SARS-CoV-2, where the similar alignment position had different amino acid residues, but by no means this is a substitution. Furthermore, the author did not mention which position alignment they used.

Response:

Thank you for pointing this out to us. We admit that the use of the terminology “substitution” and the way of representing the residue change with, for example, T444K, is inappropriate for the two evolutionarily unrelated RBDs. However, for purpose of simplicity and convenience in description, we still use the way of representing residue mutation (e.g., T444K) between evolutionarily related sequences but highlight that such representation merely represents residue change at the structurally equivalent position as shown in the structure-based sequence alignment (Figure 1D). The residue labels in Figure 4B have been modified as RBDCoV residue/RBDCoV2 residue plus the residue number of RBDCoV2 (e.g., K/R346) for differential residues at the structurally equivalent position, the term “residue substitution” has been replaced with “residue change” throughout the manuscript, and the alignment position used has been clearly pointed out to be according to RBDCoV2 numbering in the revised manuscript.

The discussion between line 704 to line 710: The authors only used binding free energy to conclude the results. However, physical evidence such as the structural change and the binding angle between RBD and ACE2 will enhance their conclusion.

Response:

The discussion mentioned by the reviewer focuses on interpreting why the RBD interface, rather than RBM, determines the higher ACE2-binding affinity of RBDCoV2 than of RBDCoV. The reasons for this, including the differences in the physical interactions and net charge between the RBD interface and RBM, have been rationalized in the manuscript.

Minor:

Keywords: “SARS-CoV-2 variants” – there is no variants in the study. “amino acid residue substitutions “ – this is a wild-type SARS-CoV and SARS-CoV-2 study, there are no amino acid residue substitutions

Response:

SARS-CoV-2 variants was removed, amino acid residue substitutions was replaced with amino acid residue changes, and an additional keyword binding interfaces was added.

Line 304 – please clarified “single joined equilibrium trajectory of each complex”, does this mean only a single model was used to calculate FEL?

Response:

We have rephrased this sentence as “For each complex, PCA analysis was performed on its single joined equilibrium trajectory to extract the most important PCs/eigenvectors.”. The approach of how the equilibrated portion of each of the 10 replicas was concatenated into a single joined equilibrium trajectory was described in the section 2.3. Yes, for each complex, its FEL was obtained based on the single joined equilibrium trajectory.

Figure 4C – the bottom x-axis had different font.

Response:

Thank you for your careful inspection. This has been modified to keep fonts consistent.

Figure 5 – The node size and the hydrogen line should have a scale. Furthermore, figure 5C, the text is not aligned, what does OG, OD1, OD2 mean?

Response:

The scale has been added to Figures 5A, B, and C and the text in Figure 5C has been aligned. OG, OD1, and OD2 in Figure 5C represent atom names in PDB format, which correspond to Og, Od1, and Od1 according to IUPAC nomenclature for standard amino acid.

Line 478 – Hydrogen bond is a dipole-dipole interaction, therefore should not use the term of donor or acceptor as there is no transfer of proton or electron.

Response:

Traditionally, when a hydrogen bond (HB) is formed, a more electronegative atom or group that is covalently bound to a hydrogen atom is called the HB’s donor, and another electronegative atom bearing a lone pair of electrons is called the HB’s acceptor (for details, see https://en.wikipedia.org/wiki/Hydrogen_bond). Anyway, we have rephrased the sentence as “… when the distance between the donor and acceptor atoms is less than 3.5 Å and the angle from the donor atom over the hydrogen to the acceptor atom is greater than 120°”.

Line 767 – This is not a residue substitution and the study did not give evidence of conformation change.

Response:

Thanks you for your careful reading, the sentence has been rewritten as “… caused by RBD residue changes can also largely impact …”.

Reviewer 3 Report

Zhang et al. successfully apply an MD simulation and molecular mechanic characterization of the interaction between the RBD of SARS-CoV-2 and its specific receptor ACE2; this interaction is then compared to that between the RBD of SARS-CoV and ACE2. In their analysis some interesting insight of this molecular recognition are reported, such as the dominant role of the residues that belong to the interface RBD-ACE2. The methodology applied in this analysis is 'solid' and interesting, even if some steps require a clearer explaination.

Unfortunately, the presentation of the results (text) in my opinion require strong improvements in term of clarity and length and concision of the whole text and abstract.

My suggestions are the following.

  1. The presentation of the results must be improved in the whole text. The sentences are too long and difficult to read and understand. Plese improve it significantly in accord to the English grammar.
  2. Please add some details about the reconstruction of the Free Energy Landscape.
  3. The paragraph 3.4 include interesting results, but is too long. Please reduce it significantly, improve the clarity of the text.
  4. In my opinion the use of 'electronegative' and 'electropositive' to indicate an object characterized by negative or positive electrostatic charge, respectively, is wrong. Electronegativity in chemistry is a different property

Author Response

Response to Reviewer 3 Comments

Reviewer Report 3

Zhang et al. successfully apply an MD simulation and molecular mechanic characterization of the interaction between the RBD of SARS-CoV-2 and its specific receptor ACE2; this interaction is then compared to that between the RBD of SARS-CoV and ACE2. In their analysis some interesting insight of this molecular recognition are reported, such as the dominant role of the residues that belong to the interface RBD-ACE2. The methodology applied in this analysis is 'solid' and interesting, even if some steps require a clearer explaination.

Unfortunately, the presentation of the results (text) in my opinion require strong improvements in term of clarity and length and concision of the whole text and abstract.

My suggestions are the following.

The presentation of the results must be improved in the whole text. The sentences are too long and difficult to read and understand. Plese improve it significantly in accord to the English grammar.

Response:

We are sorry for this. We have made an attempt to improve the presentation of our findings through extensive language editing. Many long sentences were reduced to the short sentences, some long-winded but irrelevant descriptions were removed, and the length of the paper was also reduced.

Please add some details about the reconstruction of the Free Energy Landscape.

Response:

The FEL of each complex was reconstructed using the projection of the single joined equilibrium trajectory onto the first two eigenvectors as the reaction coordinates. We have supplemented the Python script used for FEL construction in Supplementary materials as File S1.

The paragraph 3.4 include interesting results, but is too long. Please reduce it significantly, improve the clarity of the text.

Response:

Thank your for your affirming words. An attempt has been made to reduce this part and improve the clarity of the text.

In my opinion the use of 'electronegative' and 'electropositive' to indicate an object characterized by negative or positive electrostatic charge, respectively, is wrong. Electronegativity in chemistry is a different property

Response:

We are sorry for the misuse of the above terminologies. We have replaced “electronegative potential” and “electropositive potential” with “negative electrostatic potential” and “positive electrostatic potential”, respectively.

Reviewer 4 Report

The presented manuscript was really enjoyable to read and although simple bioinformatics analysis but it has a very significant influence on understanding to some extent the origin of the SARS-CoV virus. Some comments for the authors below to consider

  1. Please revise the sentence lines 27-32; 67-77;  for clarity and readability
  2. please be consistent with one space rather than two spaces after the punctuation marks
  3. In table 1 could the authors explain the calculated values between brackets and indicate in the figure legend the meaning of the values
  4. Authors need to update the information about the ORF accessory proteins and their host interaction information, which would be essential to understanding the biological importance of SARS-Cov-2 accessory proteins.
  5. I think it would be good for the manuscript to include the crystal structure of the SARS-CoV-2, SARS-CoV chimeric RBD complexed with ACE2
  6. Could the authors comment on the  N501Y mutation, which has appeared in multiple lineages, lies within the RBD, and is believed to increases its affinity for ACE2. 
  7. Is it feasible to get a table with  koff, kon, calculated KD, and equilibrium KD values for all RBD variants binding all ACE2 variants
  8. Is it doable to include the sensorgram effect of RBD mutations on the affinity and kinetics of binding to ACE2? 

Author Response

Response to Reviewer 4 Comments

Reviewer Report 4

The presented manuscript was really enjoyable to read and although simple bioinformatics analysis but it has a very significant influence on understanding to some extent the origin of the SARS-CoV virus. Some comments for the authors below to consider

Please revise the sentence lines 27-32; 67-77; for clarity and readability

Response:

The above mentioned sentences have been rewritten to convey clearly what we want to say.

Lines 27-32: “Comprehensive comparative analyses of the residue binding free energy components and IRCNs between the two complexes reveal that it is the residue changes at the RBD interface that lead to the overall stronger inter-protein electrostatic attractive force in RBDCoV2-ACE2, which not only tightens the interface packing and suppresses the dynamics of RBDCoV2-ACE2, but also enhances the ACE2-binding affinity of RBDCoV2.”.

Lines 67-77: “In the closed conformation, the ACE2-binding surface (i.e., partial surface of the receptor binding motif (RBM)) on RBDs is buried inside the trimer, inaccessible to ACE2 because of the down orientations of the three RBDs and their tight packing against one another [17]. The closed state can spontaneously convert to the open states through a hinge-like motion that progressively lifts RBDs up, thus allowing binding to ACE2 due to the full exposure of RBM [22]. Upon binding, the first ACE2-bound RBD is stabilized in the up orientation, and this promotes the other two RBDs to consecutively lift up and bind to ACE2 until reaching the fully open, three-ACE2-bound conformation, which is responsible for priming spike trimer for S2 unsheathing and the following membrane fusion [18].”.

please be consistent with one space rather than two spaces after the punctuation marks

Response:

Thank your for your careful attention to this. We have checked and fixed this issue.

In table 1 could the authors explain the calculated values between brackets and indicate in the figure legend the meaning of the values

Response:

We have revised Table 1 title as “Average values and corresponding standard deviations (SDs; shown in parentheses) of various MM-PBSA energy terms (kJ/mol) of the RBDCoV-ACE2 and RBDCoV2-ACE2 complexes calculated over their respective 100 representative structures.”. The meaning of SDs was explained in the text: “Interestingly, RBDCoV-ACE2 has larger SDs for all the energy terms than RBDCoV2-ACE2, meaning a larger dispersion around the respective energy average values and, hence, less tight inter-protein association in RBDCoV-ACE2, in agreement with earlier comparative analysis in terms of RMSD.”.

Authors need to update the information about the ORF accessory proteins and their host interaction information, which would be essential to understanding the biological importance of SARS-Cov-2 accessory proteins.

Response:

The current manuscript mainly focuses on elucidating the mechanistic reasons underlying the higher binding affinity of SARS-CoV-2 RBD to human ACE2 than that of SARS-CoV RBD to ACE2. Although the accessory proteins were shown to play critical roles in SARS-CoV-2 interaction with the host and hence in viral pathogenesis (Redondo et al., Frontiers in immunology, 2021, 12, 708264), introduction and discussion of the accessory proteins is beyond the scope of our paper.

I think it would be good for the manuscript to include the crystal structure of the SARS-CoV-2, SARS-CoV chimeric RBD complexed with ACE2

Response:

Thank you for your reminder. In the above mentioned complex crystal structure, the chimeric RBD consists of the core from the SARS-CoV RBD and RBM from SARS-CoV-2, in which a short loop (residues 438-446 ) from SARS-CoV RBM was retained (Shang et al., Nature, 2020, 581, 221–224). Therefore, this structure was not chosen as an object in our study. However, we have cited this paper in the Discussion section as experimental evidence to support our simulation-based conclusion.

Could the authors comment on the  N501Y mutation, which has appeared in multiple lineages, lies within the RBD, and is believed to increases its affinity for ACE2.

Response:

The amino acid residues at positive 501 are Thr (T) in SARS-CoV RBD and Asn (N) in SARS-CoV-2 RBD. Our results show that T501N increases the node size (Figures 5A and B) and contributes to enhancing the ACE2 binding affinity by -3.8 kJ/mol (Table S1). However, we cannot comment on the effect of N501Y on the ACE2 affinity according to our calculation results because we do not perform binding free energy calculations on RBDs of SARS-CoV-2 variants versus ACE2. Anyway, previous experimental and theoretical studies (Han, et al., Nature Communications, 2021, 12, 6103; Barton et al., Elife, 2021, 10, e70658; Shahhosseini et al., Microorganisms, 2021, 9, 926) show that N501Y indeed greatly enhances RBD binding to ACE2.

Is it feasible to get a table with  koff, kon, calculated KD, and equilibrium KD values for all RBD variants binding all ACE2 variants

Response:

A Table like this has been obtained in a previous experimental study by Barton et al. (Elife, 2021, 10, e70658), with the binding kinetics parameters and affinity measured using the surface plasmon resonance assay, although the values for the RBD of Omicron variant binding all ACE2 variants were not included in the table. Since the calculated binding free energy values in our study are merely the ones of the relative binding energies (which are only used to compare the relative difference in binding affinities of RBDCoV and RBDCoV2 to the same wild-type ACE2), the conversion of them into dissociation constant (KD) values is meaningless. Moreover, we did not calculate the binding free energy values for RBD variants versus ACE2 variants. Therefore, we cannot provide the table as suggested by the reviewer.

Is it doable to include the sensorgram effect of RBD mutations on the affinity and kinetics of binding to ACE2?

Response:

We did not perform surface plasmon resonance experiments and therefore cannot provide the sensorgram effect of RBD mutations on the affinity and kinetics of binding to ACE2.

Round 2

Reviewer 1 Report

The authors have answered all my questions and performed all attended suggestions.

Author Response

Response to Reviewer 1 Comments

Reviewer Report 1

The authors have answered all my questions and performed all attended suggestions.

Response:

Thanks.

Reviewer 2 Report

The revision has satisfactorily addressed most of my questions. I recommend the revised study to be published in Cells.

Author Response

Response to Reviewer 2 Comments

Reviewer 2 report

The revision has satisfactorily addressed most of my questions. I recommend the revised study to be published in Cells.

Response:

Thanks.

Reviewer 3 Report

Unfortunately, the new version of the manuscript does not show significant improvement in accord to my previous suggestion.

The presentation of results is low and require strong revision in term of synthesis and clarity; also the English text require strong revision.

Author Response

Response to Reviewer 3 Comments

Reviewer 3 report

Unfortunately, the new version of the manuscript does not show significant improvement in accord to my previous suggestion. The presentation of results is low and require strong revision in term of synthesis and clarity; also the English text require strong revision.

Response:

We would like to thank you for your rigorous review and constructive suggestions, which has significantly improved the presentation of our manuscript. The followings are the previous comments and our new responses.

Zhang et al. successfully apply an MD simulation and molecular mechanic characterization of the interaction between the RBD of SARS-CoV-2 and its specific receptor ACE2; this interaction is then compared to that between the RBD of SARS-CoV and ACE2. In their analysis some interesting insight of this molecular recognition are reported, such as the dominant role of the residues that belong to the interface RBD-ACE2. The methodology applied in this analysis is 'solid' and interesting, even if some steps require a clearer explaination.

Unfortunately, the presentation of the results (text) in my opinion require strong improvements in term of clarity and length and concision of the whole text and abstract.

My suggestions are the following.

The presentation of the results must be improved in the whole text. The sentences are too long and difficult to read and understand. Plese improve it significantly in accord to the English grammar.

Response:

We are very sorry for this and inconvenience it caused in your reading. The manuscript has been thoroughly revised and edited by a native English speaker, and the typos and grammar errors we found have been corrected. In order to achieve the clarity and concision, the long sentences were broken into shorter phrases when necessary, the long-winded but irrelevant descriptions were removed, and the length of the abstract and whole text was reduced to a certain extent. The presentation of the results was improved to make the article more understandable, including fixes to defects in Figure 4B and Figures 5D and E, and optimizing the description and interpretation of the results.

Please add some details about the reconstruction of the Free Energy Landscape.

Response:

The FEL of each complex was reconstructed using the projection of the single joined equilibrium trajectory onto the first two eigenvectors as the reaction coordinates. We have supplemented the Python script used for FEL construction in Supplementary materials as File S1.

The paragraph 3.4 include interesting results, but is too long. Please reduce it significantly, improve the clarity of the text.

Response:

In the previous revised version, we significantly cut and revised the Section 3.4. In the current version, we further revised and summarized this section to convey clearly what has been found.

In my opinion the use of 'electronegative' and 'electropositive' to indicate an object characterized by negative or positive electrostatic charge, respectively, is wrong. Electronegativity in chemistry is a different property

Response:

We are sorry for the misuse of the above terminologies. We have replaced “electronegative potential” and “electropositive potential” with “negative electrostatic potential” and “positive electrostatic potential”, respectively, in the revised manuscript. The terms “electronegativity” and “electropositivity” were no longer used in the revised manuscript.